# Physicochemical Properties of Lipoproteins Assessed by Nuclear Magnetic Resonance as a Predictor of Premature Cardiovascular Disease. PRESARV-SEA Study

**DOI:** 10.3390/jcm10071379

**Published:** 2021-03-29

**Authors:** Bárbara Fernández-Cidón, Beatriz Candás-Estébanez, Miriam Gil-Serret, Núria Amigó, Emili Corbella, M. Ángeles Rodríguez-Sánchez, Ariadna Padró-Miquel, Carlos Brotons, Antonio Hernández-Mijares, Pilar Calmarza, Estibaliz Jarauta, Angel J. Brea, Marta Mauri, Carlos Guijarro, Àlex Vila, Pedro Valdivielso, Xavier Corbella, Xavier Pintó

**Affiliations:** 1Bioquímica Especial y Biología Molecular, Laboratori Clínic, Hospital Universitario de Bellvitge, 08907 L’Hospitalet de Ll., Spain; barbara.fernandez@bellvitgehospital.cat (B.F.-C.); BCandas@scias.com (B.C.-E.); apadro@bellvitgehospital.cat (A.P.-M.); 2Clinical Laboratory, SCIAS-Hospital de Barcelona, 08034 Barcelona, Spain; 3Biosfer Teslab, 43204 Reus, Spain; mgil@biosferteslab.com (M.G.-S.); namigo@biosferteslab.com (N.A.); 4CIBERDEM, Universidad Rovira i Virgili, 43002 Tarragona, Spain; 5Unidad de Lípidos y Riesgo Vascular, Servicio de Medicina Interna, Hospital Universitario de Bellvitge-IDIBELL, 08907 Barcelona, Spain; emilic@bellvitgehospital.cat (E.C.); mrodriguezsa@bellvitgehospital.cat (M.Á.R.-S.); xcorbella@bellvitgehospital.cat (X.C.); 6Centro de Investigación Biomédica en Red, Fisiopatologia de la Obesidad y Nutrición CIBEROBN), Instituto de Salud Carlos III, 28029 Madrid, Spain; 7EAP Sardenya, Instituto de Investigaciones Biomédicas Sant Pau, 08025 Barcelona, Spain; cbrotons@eapsardenya.cat; 8Hospital Universitario Dr. Peset, 46017 Valencia, Spain; hernandez_antmij@gva.es; 9Servicio de Bioquímica Clínica, Hospital Universitario Miguel Servet, CIBERCV IIS Aragón, Universidad de Zaragoza, 50009 Zaragoza, Spain; mpcalmarza@salud.aragon.es (P.C.); estijarauta@gmail.com (E.J.); 10Hospital General San Pedro, 26006 Logroño, Spain; abrea@riojasalud.es; 11Hospital de Terrassa (Consorci Sanitari de Terrassa), 08227 Terrassa, Spain; MMauri@CST.CAT; 12Departamento de Esepecialidades Médicas y Salud Pública, Unidad de Medicina Interna, Hospital Universitario Fundación Alcorcón, Universidad Rey Juan Carlos, 28922 Alcorcón, Spain; CGuijarro@FHAlcorcon.es; 13Hospital de Figueres (Fundació Salut Empordà), 17600 Figueres, Spain; avila@salutemporda.cat; 14Hospital Universitario Virgen de la Victoria, IBIMA, Universidad de Málaga, 29010 Málaga, Spain; valdivielso@uma.es; 15Facultad de Medicina y Ciencias de la Salud, Universitat Internacional de Catalunya, 08017 Barcelona, Spain; 16Department of Medicine, Campus Bellvitge, Universidad de Barcelona, 08907 L’Hospitalet de Ll., Spain

**Keywords:** NMR analysis of lipoproteins, lipid profile, premature cardiovascular disease, residual cardiovascular risk, small dense LDL, lipoprotein particle number, lipoprotein precipitation

## Abstract

Some lipoprotein disorders related to the residual risk of premature cardiovascular disease (PCVD) are not detected by the conventional lipid profile. In this case-control study, the predictive power of PCVD of serum sdLDL-C, measured using a lipoprotein precipitation method, and of the physicochemical properties of serum lipoproteins, analyzed by nuclear magnetic resonance (NMR) techniques, were evaluated. We studied a group of patients with a first PCVD event (*n* = 125) and a group of control subjects (*n* = 190). Conventional lipid profile, the size and number of Very Low Density Lipoproteins (VLDL), Low Density Lipoproteins (LDL), High Density Lipoproteins (HDL) particles, and the number of particles of their subclasses (large, medium, and small) were measured. Compared to controls, PCVD patients had lower concentrations of all LDL particles, and smaller and larger diameter of LDL and HDL particles, respectively. PCVD patients also showed higher concentrations of small dense LDL-cholesterol (sdLDL), and triglycerides (Tg) in LDL and HDL particles (HDL-Tg), and higher concentrations of large VLDL particles. Multivariate logistic regression showed that sdLDL-C, HDL-Tg, and large concentrations of LDL particles were the most powerful predictors of PCVD. A strong relationship was observed between increased HDL-Tg concentrations and PCVD. This study demonstrates that beyond the conventional lipid profile, PCVD patients have other atherogenic lipoprotein alterations that are detected by magnetic resonance imaging (MRI) analysis.

## 1. Introduction

Premature cardiovascular disease (PVCD) remains an important public health issue and dyslipidemia is the most common risk factor of this condition [1] Patients with premature cardiovascular disease (PCVD) have non-conventional risk factors, both lipid and non-lipid [2,3], which entail a residual cardiovascular risk that is usually not detected or adequately controlled [4,5]. The Progression of Early Subclinical Atherosclerosis (PESA) study demonstrated that subclinical atherosclerosis is present in individuals without traditional cardiovascular risk factors (CVRFs) and with Low Density Lipoprotein cholesterol (LDL-C) at levels currently considered normal [6]. Furthermore, in the Multi-ethnic Study of Atherosclerosis (MESA) study a lower cut-off value for LDL-C (70 mg/dL) was proposed with which some individuals still present subclinical atherosclerosis [7]. Consequently, new biomarkers to estimate residual cardiovascular risk must be considered. Among them are different alterations in the number, structure, and function of plasma lipoproteins. Extended studies of lipid metabolism have shown that individuals with an excess of low-density lipoprotein (LDL) particles have a higher cardiovascular risk, even when total cholesterol and LDL-cholesterol (LDL-C) levels are within the reference range, and this risk is higher in those with a predominance of smaller and denser LDL particles, compared to subjects with a predominance of larger and buoyant LDL particles [5,6,7,8]. Currently, total serum cholesterol concentrations, cholesterol in LDL particles, cholesterol in high-density lipoprotein (HDL) particles, and total serum triglycerides (Tg) constitute the conventional lipid profile, but this profile does not allow the detection of other lipid metabolism disorders that may predispose to PCVD [6,9]. For this reason, it is of interest to determine the role of other lipoprotein characteristics beyond cholesterol content, such as the cholesterol concentration of small dense LDL (sdLDL), which are not measured in automated clinical laboratories [10,11].

To date, the reference method for the study of lipoproteins requires a centrifugation stage, which is not usually available because of its cost and lengthy time of analysis. However, other methods have recently been developed to separate sdLDL and quantify its cholesterol content. Hirano and collaborators developed a method that provides interchangeable results with ultracentrifugation and can be easily adapted to clinical laboratories [12].

Lipoproteins and their apolipoproteins may undergo alterations that affect their structure, functionality, composition, and plasma concentrations and favor their accumulation in the arterial wall, triggering greater inflammatory and immune response and promoting atherosclerosis and its complications [6,9,10]. It has been observed that among these lipoprotein alterations, the diameter and number of particles measured by nuclear magnetic resonance (NMR) are the most relevant for the prediction of cardiovascular risk [6,13].

In the present case-control study we explored the power of the physicochemical properties of lipoproteins to predict PCVD to evaluate the residual cardiovascular risk beyond c-LDL. Our hypothesis is that the advanced lipid profile may be a new reliable biomarker of residual cardiovascular risk in patients already treated with lipid-lowering drugs and in whom LDL-C has reduced. The final purpose of this study was to demonstrate that lipoprotein size and its composition can be a better predictor of PCVD than LDL-C in patients on statin therapy who have already achieved the LDL-C goal.

## 2. Materials and Methods

### 2.1. Population

In this study a group of 125 patients who had had PCVD as the first cardiovascular event between 1 January 2014 to 31 December 2016 were included. Patients of both sexes less than 50 years of age were recruited from lipid and vascular risk units affiliated with the Spanish Arteriosclerosis Society. The study protocol was approved by the Clinical Research Ethics Committee of the Hospital de Bellvitge, and all the patients signed the informed consent. PCVD was defined as a first episode of one of the following cardiovascular events: Angina pectoris with signs of coronary obstruction documented by angiography, acute myocardial infarction, stroke, transitory ischemic attack with at least one luminal stenosis ≥50% in the carotid or subclavian arteries and revascularization of any arterial territory due to atherosclerosis. All premature cardiovascular events were confirmed with objective data or a complementary test.

Patients with poor general health conditioning a short life expectancy, intellectual impairment, kidney or liver failure, and other causes of secondary dyslipidemia were excluded.

Control subjects were selected from apparently healthy workers from a car factory paired by age and sex.

The characteristics of the two groups are described in Table 1.

### 2.2. Variables Included

Lipid variables from the conventional lipid profile were evaluated, including plasma cholesterol concentrations, Tg, LDL-C, and HDL-C. Extended lipid profile biomarkers were: sdLDL-C, very low density lipoprotein cholesterol (VLDL-C), VLDL-Tg, VLDL particle number (VLDL-P) including small (sVLDL-P), medium (mVLDL-P), and large (lVLDL-P), VLDL diameter (VLDL-Z), intermediate density lipoprotein cholesterol (IDL-C), IDL-Tg, LDL-Tg, LDL particle number (LDL-P) including small (sLDL-P), medium (mLDL-P), and large (lLDL-P), LDL diameter (LDL-Z), HDL-Tg, HDL particle number (HDL-P) including small (sHDL-P), medium (mHDL-P), and large (lHDL-P), and HDL diameter (HDL-Z).

### 2.3. Sample Collection

PCVD group samples: Venous blood samples were withdrawn after a 12-h overnight fast. All serum samples were collected in tubes without anticoagulant and with separating gel. Serum samples were conserved at −80 °C until analysis.

Control group samples: Venous blood samples were withdrawn after a 12-h overnight fast. Plasma was collected in tubes containing the anticoagulant EDTA-K3 (BD, Vacutainer, NJ, US) and were centrifuged immediately for 15 min at 1500 g and at 4 °C. The plasma was immediately separated and stored at −80 °C until biochemical and NMR analysis.

### 2.4. Methods

#### 2.4.1. Lipoprotein Precipitation Techniques

sdLDL-C concentrations were obtained using a specific lipoprotein precipitation method previously described by our group and adapted for use in routine clinical laboratories. G centrifugation force and cooling samples temperature were modified in order to improve lipoprotein precipitation with a density <1.044 g/mL in hypertriglyceridemic samples [14].

#### 2.4.2. Nuclear Magnetic Resonance (NMR) Analysis

Before 1H-NMR analysis, 200 μL of serum were diluted with 50 µL deuterated water and 300 µL of 50 mM phosphate buffer solution (PBS) at pH 7.4. 1H-NMR spectra were recorded at 310 K on a Bruker Avance III 600 spectrometer (Bruker BioSciences Española S.A., Rivas Vaciamadrid, Madrid, Spain) operating at a proton frequency of 600.20 MHz (14.1 T) as previously reported [15].

The Liposcale^®^ test was used to obtain the whole lipoprotein profile including lipid concentrations, Tg and cholesterol, size and particle number of three VLDL, LDL, and HDL, as well as the particle number of nine subclasses (large, medium, and small VLDL, LDL, and HDL). Particle concentrations and diameter were obtained from the measured amplitudes of the methyl group NMR signals that differ among the lipoprotein subclasses [15].

### 2.5. Statistical Analysis

Descriptive analysis of the advanced lipoprotein panel of the two groups was performed. Results are shown as medians and 25th and 75th percentiles and comparative analyses of the variable medians between groups were performed using the Mann–Whitney U test.

The statistical approach consisted in different steps:

First, a logistic univariate regression analysis was carried out to explain the contribution between each variable and the development of PCVD. The results of each variable and its *p*-value are shown in Appendix A.

Secondly, Spearman correlation rho were calculated for variables with *p* < 0.05 in the univariate analysis that were significantly related to PCVD.

Among the pairs of variables presenting rho > 0.5, the variable with the lowest significance in the univariate analysis was not considered for the final model.

The model selected is the one which presents the highest goodness-of-fit evaluated with the most significant Hosmer–Lesmeshow index (HL) and the highest diagnostic accuracy calculated by receiver operating curve (ROC) analysis. The *p*-value and pseudo-R^2^ de Nagelkerke of the model were estimated and odds ratios (OR) of each variable included in the model were calculated.

In addition, ROC analysis was performed. The area under the curve (AUC), and the sensitivity and specificity of the model were estimated in order to establish the efficiency of the model for PCVD prediction. Contributions of each variable to the risk of presenting a premature ischemic event were evaluated taking into account the OR results obtained in the logistic regression. The ROC analysis was also cross validated to avoid the risk of over-fitting and to increase the predictive capacity using the Venetian blinds method.

Stata^®^ 14 software (StataCorp LLC, College Station, TX, USA) was used to carry out the statistical analyses.

## 3. Results

Table 2 shows the advanced lipoprotein data of the two groups. Patients with PCVD presented lower concentrations of large, medium, and small sized LDL particles compared to controls. In the PCVD group, the LDL particle diameter was smaller and sdLDL-C and LDL-Tg concentrations were higher. Large-VLDL particles concentration was also higher in patients with PCVD. HDL-Tg concentrations were higher and had a larger diameter in PCVD patients compared to controls. Only variables showing significant differences between groups were evaluated for inclusion in the multivariate logistic regression model. After that the collinearity study was carried out. The variables excluded were those that correlated with HDL-Tg (*p* = 1.35 × 10^−25^) as follows: lHLDL-P (*p* = 3.25 × 10^−11^) rho = 0.471; mHLDL-P (*p* = 2.95 × 10^−7^) rho = 0.537; sHDL-P (*p* = 0.001) rho = 0.157; HDL-Z (*p* = 1.30 × 10^−10^) rho = 0.199; lVLDL-P (0.001) rho= 0,654; LDL-P (*p* = 7.31 × 10^−13^) rho = 0.385; mLDL-P (*p* = 2.41 × 10^−13^) rho = 0.485; sLDL-P (*p* = 1.30 × 10^−8^) rho = 0.272; LDL-Z (*p* = 1.02 × 10^−9^) rho = 0.267; LDL-C (*p* = 2.90 × 10^−13^) rho = 0.469.

Variables that were candidates for the final model were: sdLDL-C, lLDL-P, HDL-Tg, VLDL-Z, LDL-Tg. According to the highest goodness-of-fit and the highest diagnostic accuracy calculated by a ROC curve analysis, the model selected included sdLDL-C, lLDL-P, HDL-Tg.

After discarding covariances, the most powerful biomarkers that could contribute to predicting PCVD were sdLDL-C, HDL-Tg, and lLDL-P. Table 3 shows the results of the multivariate logistic regression analysis and the ORs of the model. Logistic regression was statistically significant with a *p* < 0.001 and a pseudo-R^2^ of Nagelkerke = 0.73.

The ORs of each variable indicate the risk of presenting a premature ischemic event caused by this variable. All variables included in the logistic regression contributed significantly to the risk of PCVD with a *p* < 0.05. A strong relationship was observed between the increase in HDL-Tg concentrations and PCVD.

The diagnostic accuracy of the model to predict PCVD was estimated using ROC analysis (Figure 1). The AUC obtained and its 95% confidence interval (CI) was 0.95 (0.92–0.99). The sensitivity was 94.7% and the specificity was 80.3%. The ROC analysis was cross-validated to avoid the risk of over-fitting and to increase the predictive capacity using the Venetian blinds method. The AUC remained unaltered (<1%) after the cross-validation process.

## 4. Discussion

There is a need to characterize the residual risk of PCVD patients without dyslipidemia in the conventional lipid profile [16,17]. Several studies have shown that obtaining the lipoprotein profile by NMR in these patients could help to predict the risk of cardiovascular disease [18,19,20]. Although the independence of NMR lipid variables versus traditional lipid profile has not been demonstrated, there are some studies that support the superiority of advanced lipid profile in estimating cardiovascular risk [21,22,23]. In this prospective observational study, we evaluated the contribution of advanced lipid profile parameters in a multivariate logistic regression model for PCVD prediction. NMR study of the lipoprotein profile (Liposcale^®^ test) consists in the mathematical deconvolution of the various resonance signals given by the methyl groups of the lipoprotein lipid nucleus [16]. The Liposcale^®^ test analyzes the average diameter, cholesterol and Tg content, and concentrations of the number of VLDL, LDL, and HDL lipoproteins, and the content of Tg and cholesterol of IDL particles and the concentration of the number of VLDL, LDL, and HDL lipoprotein subclasses (large, medium, and small).

We found significant differences between groups in the different subclasses of lipoproteins, including their diameter and composition. LDL-P and LDL-C were significantly lower in PCVD patients than in control subjects which may be the consequence of a higher use and intensity of lipid-lowering treatments in PCVD patients. However, it should be noted that neither of these two variables were included in the predictive model.

Particle diameter is one of the main lipoprotein characteristics that influences atherogenicity. It has been observed that smaller LDL and VLDL are more atherogenic because they are more easily retained in the endothelium [24,25]. However, the exception to this are large buoyant VLDL (lVLDL-P) which are more atherogenic since their metabolism generates sdLDL particles that are more atherogenic than large buoyant LDL particles which are formed from small VLDL particles. In the present study, PCVD patients had higher concentrations of lVLD-P and sdLDL particles, and sdLDL-C than control subjects.

It is well known that cholesterol contained in small LDL particles is more atherogenic, since it is directly related to the thickening of the intima media [9,19] and to a higher risk of cardiovascular disease. However, there is also evidence that in some patients with coronary disease large LDL particles predominate compared to other LDL subclasses. In addition, it has been observed that large LDL is an independent determinant of carotid intima media thickness (cIMT) in healthy men, and in the MESA study [6] both large LDL and small LDL particle numbers correlated with cIMT.

In the multivariate logistic regression model of the present study only three biomarkers were significantly associated with PCVD: sdLDL-C, HDL-Tg, and large LDL-P. These variables explained 72% of PCVD risk. The role of HDL-Tg in this model is especially relevant because an increase of 1 mg/dL in serum HDL-Tg was associated with an increase in PCVD risk of 62%. On the other hand, an increase of 1 mmol/L in the number of large LDL particles was associated with a decrease of 4.4% in PCDV risk, whereas the effect of sdLDL-C on PCVD risk, although significant, was small. This model presented a high AUC with good sensitivity suggesting that it could be used for PCVD risk screening, especially in subjects with first-degree relatives with cardiovascular disease who have low LDL concentrations, or in patients with cardiovascular disease that is not explained by traditional biomarkers and in whom additional residual risk must be considered. These data agree with those from other studies and support the contention that NMR lipoprofile biomarkers are stronger predictors of PCVD [26] and subclinical atherosclerosis [27] than other conventional risk factors such as LDL-C concentrations.

The role of HDL-Tg as a predictor of PCVD risk may be explained by the decrease in the antiatherogenic properties of HDL that has been observed as these particles become enriched in Tg [28,29].

On the other hand, the value of HDL particle diameter as a predictive factor of CVD is controversial. Some studies demonstrate that HDL-P and SHDL-P are inversely related to CVD [30,31] and to all-cause as well as cardiovascular mortality in CAD patients [32]. In our study patients in the PCVD group had a higher number of large HDL particles and a smaller number of small HDL particles than the control group.

The number of LDL particles can also be estimated quantifying plasma Apo B concentrations, since every LDL particle contains only one Apo B molecule. Some studies have shown that Apo B is a more powerful marker of cardiovascular disease risk than LDL-c and non HDL-C [33]. In addition, in other studies the increased atherogenic effect of small LDL particles has not been observed, and the number of both total LDL and HDL particles has been described as a better predictor of cardiovascular risk. In fact, it has been reported that a percentage greater than 65% of the interindividual differences in cardiovascular risk is due to the number of lipoprotein particles and not to the diameter [34].

The discordance between LDL-C and advanced lipid profile data is mainly observed in individuals at risk of cardiovascular disease presenting high Tg concentrations, type 2 diabetes mellitus or metabolic syndrome (MS) [35]. MS patients present the highest prevalence of Tg, sdLDL particles, and large HDL particles.

The results of this study show that beyond the conventional lipid profile, the NMR advanced lipid profile can detect other lipoprotein alterations related to the residual risk of PCVD. Some clinical guidelines recommend the measurement of LDL-P concentrations in the follow-up of patients with MS [36] and type 2 diabetes mellitus [37]. Some expert panels and international guidelines propose to guide lipid-lowering therapy based on LDL-P concentrations instead of LDL-C concentrations, establishing a discriminant value of LDL-P <1000 nmol/L [38].

As we have hypothesized, PCVD individuals that were on statin therapy, and whose LDL-C values were lower than those of control subjects, had a higher concentration of sdLDL-C. These results support the relevance of LDL cholesterol composition and size on PCVD progression and its potential as a new biomarker for residual cardiovascular risk.

This study has several limitations. Lp(a) measures were not considered in this study as the control group was recruited in primary care centers where this determination is not available. Lastly, these hypothesis-generating findings warrant further studies as no independent validation has been done.

## 5. Conclusions

Beyond the conventional lipid profile, PCVD patients have other atherogenic lipoprotein alterations that are not detected by the conventional lipid profile. The study of lipoproteins by NMR may allow the detection of these alterations and a broader understanding of the vascular risk related to dyslipidemias.

## Figures and Tables

**Figure 1 jcm-10-01379-f001:**
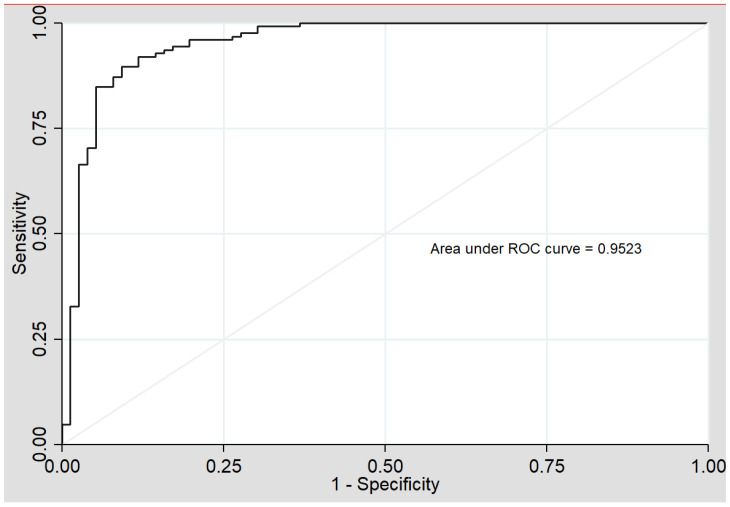
Model 1 ROC (receiver operating curve) analysis (sdLDL-C, HDL-Tg, lLDL-P). AUC (area under the curve).

**Table 1 jcm-10-01379-t001:** Clinical data of patients and control subjects.

Variables (Units)	Control Group *n* = 190	PCVD Group *n* = 125
Age (years)	47 (3.7)	46.5 (4.4)
Male (%)	150 (78.9%)	99 (79.2%)
Coronary artery disease	-	79 (63.2%)
Stroke or TIA	-	46 (36.8%) ^a^
BMI (kg/m^2^)	26.7(3.6)	29 (4.98) ^a^
Diabetes mellitus	3 (1.6%)	16 (12.8%) ^a^
Hypertension	22 (11.6%)	48 (38.4%)
Smoking	59 (31%)	32 (25.6%)
Lipid-lowering drugs	20 (10.5%)	116 (92.8%) ^a^
Anti-platelet drugs	3 (1.6%)	118 (94.4%) ^a^
Cholesterol (mg/dL)	207.4 (33.3)	155.3 (40.8) ^a^
HDL-Cholesterol (mg/dL)	56.8 (12.4)	44.5 (14.8) ^a^
Triglycerides (mg/dL)	121.1 (98.6)	140.6 (99.6)
LDL-C (mg/dL)	144.38 (12.5)	96.49 (7.9) ^a^

Data are expressed as mean (standard deviation) or case number (percentage). PCVD: Premature cardiovascular disease; TIA: Transitory ischemic attack; SBP: Systolic blood pressure; HDL-Cholesterol: High-density lipoprotein cholesterol, LDL-C: Low density lipoprotein-cholesterol. (^a^) indicates differences between groups with *p* < 0.05.

**Table 2 jcm-10-01379-t002:** Comparison between the advanced lipoprotein panels of the two groups.

Variables (Units)	Control Group (*n* = 190)	PCVD Group (*n* = 125)	*p* Value
Very Low Density Lipoprotein			
Cholesterol (mg/dL)	12.27 (6.90–21.16)	14.21 (8.44–19.94)	0.173
Triglycerides (mg/dL)	56.72 (33.74–82.68)	61.15 (43.67–92.71)	0.06
Total particles (nmol/L)	41.22 (23.86–62.63)	41.57 (30.93–64.28)	0.143
Large particles (nmol/L)	0.85 (0.53–1.29)	1.27 (0.94–1.76)	<0.0001
Medium particles (nmol/L)	4.57 (3.05–6.39)	5.04 (3.21–8.13)	0.245
Small particles (nmol/L)	36.00 (20.21–54.55)	35.11 (27.05–55.86)	0.138
Diameter (nm)	42.13 (42.01–42.25)	42.28 (42.14–42.43)	<0.0001
Intermediate Density Lipoprotein			
Cholesterol (mg/dL)	8.16 (5.53–11.05)	9.07 (7.22–11.43)	0.074
Triglycerides (mg/dL)	10.05 (7.91–12.96)	9.81 (8.27–11.60)	0.683
Low Density Lipoprotein			
Cholesterol (mg/dL)	144.38 (134.08–169.37)	96.49 (83.14–112.22)	<0.0001
Triglycerides (mg/dL)	19.27 (11.61–23.86)	12.07 (9.66–15.01)	<0.0001
Total particles (nmol/L)	1457.93 (1312.05–1648.66)	994.78 (880.20–1144.72)	<0.0001
Large particles (nmol/L)	221.12 (186.67–238.65)	158.76 (139.20–175.46)	<0.0001
Medium particles (nmol/L)	519.99 (422.60–605.56)	246.81 (187.99–308.80)	<0.0001
Small particles (nmol/L)	724.35 (642.38–861.24)	587.73 (532.41–671.76)	<0.0001
Cholesterol small particles (mmol/L)	320.00 (180.15–460.06)	420.38 (265.28–704.79)	<0.0001
Diameter (nm)	21.16 (20.99–21.33)	20.91 (20.78–21.06)	<0.0001
High Density Lipoprotein			
Cholesterol (mg/dL)	48.86 (38.76–57.78)	45.47(39.92–51.72)	0.101
Triglycerides (mg/dL)	7.88 (6.07–9.85)	14.61(11.41–17.52)	<0.0001
Total particles (µmol/L)	24.78 (20.18–28.12)	24.18 (21.05–27.86)	0.855
Large particles (µmol/L)	0.23 (0.19–0.27)	0.26 (0.24–0.30)	<0.0001
Medium particles (µmol/L)	6.80 (5.66–8.69)	9.02 (7.93–10.43)	<0.0001
Small particles (µmol/L)	16.83 (13.93–19.34)	14.98 (12.08–17.57)	0.003
Diameter (µmol/L)	8.21 (8.17–8.25)	8.30 (8.24–8.34)	<0.0001

Data are expressed as medians (interquartile interval).

**Table 3 jcm-10-01379-t003:** Results of the multivariate logistic regression analysis.

Variables (Units)	OR	[CI 95%]	*p* Value	R^2^
Constant	27.367	1.650 to 453.845	0.021	0.730
sdLDL-C (mmol/L)	1.002	1.000 to 1.003	0.030	
HDL-Tg (mg/dL)	1.622	1.296 to 1.764	<0.0001	
lLDL-P (nmol/L)	0.956	0.941 to 0.967	<0.0001	

OR (odds ratio); [CI 95%] (95% confidence interval); R^2^ (pseudo-R^2^ of Nagelkerke), sdLDL-C (small dense LDL-cholesterol); HDL-Tg (HDL-triglyceride); lLDL-P (large LDL particle number).

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
