# Peer review of "Physicochemical Properties of Lipoproteins Assessed by Nuclear Magnetic Resonance as a Predictor of Premature Cardiovascular Disease. PRESARV-SEA Study"

_jcm, 2021, doi:10.3390/jcm10071379_

Round 1

Reviewer 1 Report

The study by Fernandez-Cidon investigates physicochemical properties of lipoproteins in relation to premature CVD risk. The study aims to provide risk factors to detect residual risk in a PCVD cohort, which cannot be explained by traditional risk factors.

  1. The authors should also evaluate the OR for known risk factors, to judge the benefit of introducing a novel factor. Therefore, the analysis should include OR for LDL-C, TC, waist circumference and so on. This is important, since Mora et al. 2009 Circulation has described NMR lipoprotein subfractions as CVD risk markers and came to the conclusion that they were not independent of standard lipid measures.
  2. Introduction and references: The authors should pay attention to the existing literature and the current state of literature. The introduction does not fully address the available literature and the current state of research in sufficient detail.
  3. Line 263 states: “Small HDL are independently associated with CVD and cIMT [25], whereas large HDL particles are associated with a protective effect.” This statement is not supported by the current literature. First, references 25 does not contain any data on small HDL particles. The second part of the sentence has no references at all. There is ample literature available for the connection between HDL subfractions and CVD risk. ( Albers et al. 2016, Atherosclerosis, Mora et al. 2009 Circulation, Krauss et al. 2010 COL).
  4. Table 1: Significances should be moved to the PCVD group.
  5. Why are values for LDL-cholesterol not presented?
  6. Abstract: Line 44: Sentence is misleading: “physicochemical properties of the lipoproteins was evaluated by lipoprotein precipitation and nuclear magnetic resonance (NMR) techniques”. It should rather state that lipoprotein precipitation was used for sdLDL and full serum was used for NMR analysis.
  7. Table 2: The data presentation is somewhat of confusing. Maybe the authors find a way to better organize the table to make it more reader friendly. There are examples from other studies, which have done better. For example, Mora et al. 2009 Circulation
  8. The author should better describe within the results section the reasoning and the selection criteria for the selection of sdLDL-C, HDL-Tg and iLDL-P as their prime parameters.
  9. Line 231: The statement “The atherogenic potential of lipoprotein particles is related to their diameter.” is misleading since a variety of factors determine whether a lipoprotein particle is atherogenic. The size alone does not render a lipoprotein atherogenic.
  10. Line 91 states "using disruptive -analytical approaches". What does that mean?

Reviewer 2 Report

With interest I have been reading the manuscript of ‘Fernandez-Cidon B et al. Physicochemical properties of lipoproteins assessed by nuclear 2 magnetic resonance as a predictors of premature cardiovascular 3 disease. PRESARV-SEA study’. Trying to disentangle underlying more specific disease causing factors for PCVD is very important, for a better understanding of why these patient get CVD at such a young age. I would hereby  like to address my major and minor concerns:

Major concerns:

  1. The manuscript lacks a novel vision on PCVD and why lipid profiles are interesting in that respect. The authors discus in the introduction and discussion that the smaller the particle the more atherogenic, but refute this in the discussion leaving it for the reader to decide. Could the authors better describe what they would have expected, and how, what they find influenced what they expected? This also holds true for the conclusion, which lacks a firm statement on what the authors found an how this could change our further for PCVD.
  2. The manuscript lacks a structured approach on how to statistically handle the different lipid particles and although the authors mentioned to have addressed collinearity, it is difficult to fully understand whether they succeeded. Were all variables analysed for collinearity or just some pairs as the authors mention and which pairs? Could the authors elaborated more on that.
  3. In my opinion the analysis should separate in different parts each containing different aspects of the lipid profile. A zoom in approach, in which you first analyse the predictive value of either ApoA1 or ApoB on PCVD is analysed. Second, the different parts ApoA1 and ApoB harbour, for instance the analysis for ApoB could be split in separate analysis of chylomicron, VLDL, LDL and Lpa. Finally the different parts of the different splitted groups can be taken together to reveal which is the major driver of the PCV risk. It is not clear whether ApoA1, ApoB, chylomicron and Lpa particles were also taken into account to begin with and if not, why that is.
  4. I believe some caution is needed when interpreting the ROC curve, since the model is derived from the same population. This means that you first investigate which variables best describe the risk of PCVD and secondly, show that a combination of these variables has the best sensitivity and specificity. It is more of the same. Better to cross-validate on a separate population, if possible of course.
  5. The data is heavily influenced by the fact that the PCVD uses cholesterol lowering medication. The authors mention in the discussion that particle number might be more important than LDL-c levels, but particle number is surely influenced by cholesterol lowering medication. Particle number is now even lower in the PCVD group as compared to controls according to table2. So, the variables the are used in the diagnostic accuracy model reflect treated PCVD lipid profiles and might be a better reflection of compliance than actual CV risk and can therefore not be used as a prediction tool for family members in my opinion, or am I missing something.
  6. Finally, I miss other variables such as BMI, insulin resistance markers, smoking, blood pressure ect and its relationship with the lipid profile. Was the data controlled for these items? And if not why?

Round 2

Reviewer 1 Report

The authors have addressed my comments sufficiently. I have no further comments